# HDRIVE: HDR IMAGE VISUAL EVALUATION METRIC FOR SDR TO HDR UPCONVERSION QUALITY ASSESSMENT

## ABSTRACT

HDR displays are becoming increasingly common on both TVs and mobile devices, that requires to adapt existing legacy SDR to HDR screens. Several algorithms have been developed for SDR-to-HDR upconversion, also known as Inverse Tone Mapping (ITM). However, there is still a lack of reliable metrics for assessing the quality of these algorithms. This is due in part to the ill-posed nature of the ITM task, where the most visually pleasing result can significantly differ from the original image. In this work, we propose a novel state-of-the-art no-reference video quality metric for evaluating upconverted HDR content. To support our approach, we collect a large-scale dataset of human visual preferences, capturing both the perceived visual appearance and quality of HDR videos. The HDR ITM video quality metric might be very helpful and drive the rapid advancement of SDR-to-HDR algorithms development and benchmarking. Both the metric and the training dataset are publicly available for download via the provided link.

## 1 INTRODUCTION

The number of high dynamic range (HDR) displays in consumer devices has grown rapidly in recent years, spanning both mobile devices and televisions. HDR displays are capable of performing significantly brighter, more vivid, and more contrast-rich images compared to their standard dynamic range (SDR) counterparts. However, a substantial portion of available content remains in SDR, preventing full utilization of HDR display capabilities. To address this limitation, inverse tone mapping (ITM) algorithms have been developed to convert SDR content into HDR format. In recent years, several methods, including deep learning based, have been proposed to up-convert SDR images (et al., 2017b;c; 2018a;b; Zhang & Lalonde, 2017; et al., 2020b).

The visual quality of HDR images and videos produced by ITM algorithms should be assessed to guide algorithm development and enable the selection of better solutions. Although several ITM methods have been proposed, there remain significant limitations in how their output quality is evaluated. Currently, very few publicly available benchmarks provide human-annotated subjective scores for the visual quality of several ITM algorithms et al. (2023); Voronin et al. (2024). However, it is hard to use subjective assessments at scale due to the high cost and complexity of votes collection. Therefore, there is a strong need for accurate and reliable objective visual quality metrics tailored to the evaluation of ITM produced images.

Several ITM methods are optimized using traditional image similarity metrics such as PSNR, SSIM, or perceptual metrics designed for HDR content, such as HDR-VDP et al. (2011). However, these full-reference metrics have important limitations in the context of ITM. The HDR output produced by an ITM algorithm may differ significantly from the reference HDR image due to the ill-posed nature of the task, but still appear visually pleasing to human observers. Moreover, as shown in Hanji et al. (2022), training ITM models using full-reference losses can lead to undesired behavior, such as learning to replicate the gamma characteristics of the training data instead of improving perceptual image quality.

In recent years, no-reference quality metrics have gained significant popularity across various video processing domains. However, the task of ITM quality assessment differs substantially from traditional scenarios such as compression, user-generated content (UGC), noise etc. As a result, existing

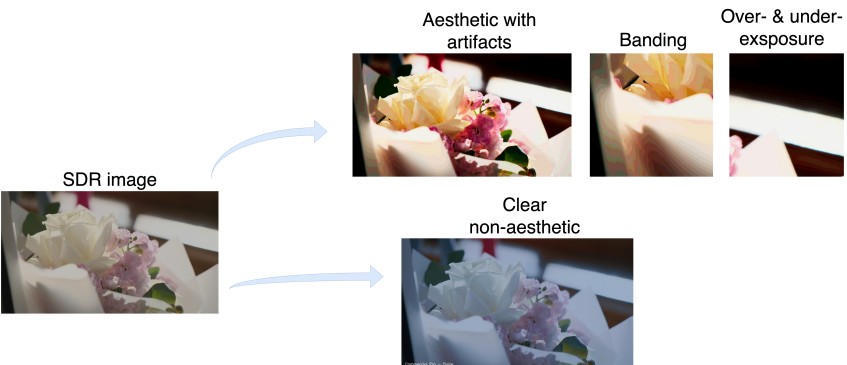

Figure 1: Illustration of the ITM quality assessment task. A robust quality metric must capture both technical distortions (e.g., banding, overexposure) and aesthetic qualities such as globally pleasant tone and color distribution

no-reference metrics often generalize poorly to the ITM domain and require specialized training to perform effectively. While an HDR-specific no-reference metric has been proposed, its correlation with subjective scores remains moderate, limiting its practical applicability. Therefore, we introduce a novel no-reference ITM quality metric specifically designed to assess the perceptual quality of HDR images produced from SDR inputs.

The reason for the lack of accurate learning-based ITM quality metrics is the limited availability of subjectively labeled training data. Supervised training of neural networks requires large-scale datasets annotated with human judgments, which are currently absent for the ITM task. To address this gap, we construct a new training dataset with human-labeled visual quality scores, specifically designed to support the development of no-reference ITM metric.

When evaluating HDR images produced by ITM algorithms, a quality metric must account for both aesthetic visual appeal for the human observer, for example general tone distribution or overall palette, t and the absence of technical artifacts. Artifacts in ITM outputs arise from the limited information content of the SDR images, including underexposed or overexposed regions, and banding artifacts resulting from insufficient signal precision. To comprehensively capture these aspects, we construct a dataset covering three key use cases: aesthetic preferences based on human judgments of tonal and color quality, technical quality, involving annotations of visible artifacts, and overall perceptual quality of the HDR output. This dataset provides the foundation for training and validating our proposed no-reference ITM quality metric.

In this paper, we propose a novel no-reference quality metric specifically designed to evaluate HDR images generated from SDR content using inverse tone mapping (ITM) algorithms. The proposed metric demonstrates state-of-the-art performance, achieving the highest correlation with human judgments on existing ITM subjective quality datasets. To enable effective training of the metric, we construct a large-scale subjective dataset tailored to the specific characteristics and challenges of ITM quality assessment. The main contributions of this work are as follows:

- A new no-reference quality metric for ITM-enhanced HDR images, capable of assessing both aesthetic fidelity and technical correctness.
- A large-scale human-labeled dataset different aspects of ITM output quality.
- A comprehensive IQA metrics evaluation for ITM task demonstrating that the proposed metric outperforms existing full- and no-reference metrics on the existing ITM benchmarks.

## 2 RELATED WORKS

Inverse tone mapping (ITM) aims to expand standard dynamic range (SDR) images to high dynamic range (HDR), enabling better rendering on HDR-capable displays. Early ITM methods focused on heuristic image-based operations. Recent advancements in ITM have leveraged deep learning. et al.

(2017b) proposed HDRCNN, a CNN-based approach to hallucinate missing highlights et al. (2017c) trained networks to generate multiple exposures from a single image for HDR reconstruction. ExpandNet et al. (2018a) and other architectures et al. (2018b); Zhang & Lalonde (2017); et al. (2020b) refined end-to-end ITM pipelines with higher visual fidelity, sometimes incorporating physical imaging priors into training.

Most traditional image quality metrics such as PSNR or SSIM require ground-truth references, which are often unavailable for HDR images generated by ITM. To address this, no-reference (NR) quality metrics have been explored. TMQI by Kundu et al. et al. (2017a) used natural scene statistics and learned regressors to evaluate tone-mapped HDR images. et al. (2020a) proposed NoR-VDPNet, a CNN trained to emulate full-reference HDR-VDP predictions. Other approaches incorporate perceptual cues specific to HDR, such as contrast masking and luminance sensitivity Deng et al. (2021); Madhusudana et al. (2022a). HDR-VDP et al. (2011) is widely regarded as a perceptually aligned full-reference metric tailored to HDR. et al. (2008) also proposed a dynamic-range-independent measure suitable for tone mapping scenarios. Variants like logPSNR or PU-PSNR have been proposed to improve upon standard PSNR by operating in perceptually uniform domains.

Over the last decade, deep learning has significantly advanced NR-IQA methods, enabling them to model complex distortions and perceptual features. These general-purpose NR-IQA models are typically trained on large-scale datasets containing human opinion scores for diverse types of distortions, such as compression, blur, noise, and transmission artifacts.

Bosse et al. (2018) proposed an early deep architecture that uses both patch-level and image-level predictions. Their model extracts features from non-overlapping patches and then aggregates them using a learned pooling strategy. This hierarchical formulation was shown to improve robustness over purely global models. Li et al. (2020b) introduced a probabilistic quality representation model that outputs a distribution over quality scores rather than a scalar. This approach better captures the uncertainty in subjective ratings and is trained using KL-divergence loss. PaQ-2-PiQ Ying et al. (2020) is another notable method that introduced a pairwise comparison-based training strategy, circumventing the need for exact MOS labels. It learns to predict a relative ranking between image pairs, using a ResNet-50 backbone and contrastive loss. This strategy is especially beneficial when subjective ground truth is noisy or inconsistent.

While these general-purpose models perform well on natural image distortions and aesthetic judgments, their effectiveness on specialized tasks such as inverse tone mapping (ITM) remains limited. The training data for such models rarely includes HDR content or SDR-to-HDR distortions. As a result, they tend to misjudge the perceptual severity of ITM-specific artifacts such as overexpansion, banding in high luminance regions, or unnatural color shifts. Moreover, most models are trained on LDR images viewed on SDR displays, lacking adaptation to the luminance range and perceptual characteristics of HDR viewing conditions.

Recent evaluations et al. (2023) have shown that applying general-purpose NR-IQA metrics to ITM outputs yields moderate correlations with human judgments at best. Voronin et al. (2024) introduce both a benchmark dataset and an ITM evaluation metric. This motivates the development of task-specific NR-IQA models that incorporate both domain-specific artifacts and viewing assumptions, as presented in our work.

## 3 DATASET

Inverse tone mapping (ITM) is an ill-posed problem, meaning that the reconstructed HDR image may significantly deviate from the original. Even when the training process uses a "ground truth" HDR image from which the corresponding SDR image was derived, the restored HDR version can differ notably from the original while still exhibiting high visual appeal. This limits the applicability of full-reference metrics, as demonstrated in [ref], optimizing for full-reference loss may lead models to reproduce training data gamma characteristics rather than enhance perceptual quality.

Existing no-reference image quality metrics are typically trained on common distortion types, which differ from the artifacts introduced during SDR-to-HDR conversion. Consequently, evaluating ITM outputs requires assessing both their visual attractiveness and their technical quality similar to approaches used for user-generated content (UGC) quality assessment in Wu et al. (2023).

To address this, we collect subjective annotations capturing two perspectives: (1) overall visual appeal, including image structure and color naturalness, and (2) perceptual discomfort or annoyance caused by specific artifacts. This section describes the preparation of our training dataset and the methodology used for subjective label collection.

Currently, there are very few datasets specifically designed for evaluating the quality of inverse tone mapping (ITM). To address this gap, we publicly release our collected dataset to support future research in this area.

An important challenge in dataset construction lies in generating diverse distorted versions of the content. Relying solely on a limited set of scientific or industrial ITM algorithms poses a significant risk: due to the small number of available methods and the high tonal diversity in real-world content, a model trained on such data may learn to rank specific algorithms rather than genuinely assess visual quality. As a result, the introduction of a previously unseen ITM method during evaluation may lead to unreliable quality predictions.

To mitigate this issue, we introduce a set of synthetic distortion techniques that manipulate color, tone, and palette properties. These transformations are applied to SDR/HDR image pairs to generate perceptually diverse variants. This strategy enables the collection of human preferences over a broader range of tone and color representations, allowing us to more reliably identify which outputs are perceived as most visually pleasing.

## 3.1 DATASET PREPARATION

We generate multiple HDR image variants with diverse tonal and color characteristics to train the metric to reflect human aesthetic preferences. As previously discussed, the goal is to produce plausible HDR renditions that could have resulted from different inverse tone mapping (ITM) algorithms. While many modern neural ITM methods attempt to reconstruct lost details in highlights or shadows, our focus is on generating aesthetically pleasing tonal representations rather than restoring physical scene content. Therefore, we employ a set of parameterized tone mapping and color manipulation functions. By sampling random parameters for each function, we produce a wide variety of perceptually distinct HDR versions from the same SDR or HDR source. These variants reflect different possible aesthetic interpretations of the scene, enabling the model to learn preference-driven quality assessment. To generate diverse distortions for subjective evaluation, we employed three classes of transformations: (1) standard local and global tone mapping operators, (2) 3D look-up tables (3D LUTs), and (3) automated color grading via tone curve manipulation. For each source image, we synthesized eight variant images by applying one or two randomly selected transformations with randomly sampled parameters.

We used a selection of local and global tone mapping algorithms, including histogram equalization, gradient relighting, CLAHE, AkyuzEO, MasiaEO, HuoPhysEO, and KovaleskiEO. These operators were modified to allow greater parametric flexibility, producing a wide range of results depending on input settings.

Over 200 3D LUTs were incorporated, sourced from publicly available color grading forums and stock repositories under appropriate licenses. These LUTs simulate various aesthetic styles and color transformations used in professional post-processing.

To further diversify color and contrast variations, we synthesized random tone curves by sampling control points and fitting monotonic functions through them. For each RGB channel, we independently selected four input and four output control points to construct a mapping curve. This process enables a wide range of channel-specific contrast adjustments: for example, compressing mid-tones while expanding highlights, or darkening shadows while preserving mid-tone detail. Randomized control point selection ensures diversity while preserving physical plausibility. Figure 2 shows the transform scheme.

Together, these methods create a rich and perceptually varied set of image distortions suitable for training human-preference-aware quality metrics. At this stage, we do not intentionally introduce synthetic artifacts, as we concentrate on overall image aesthetics. However, some common artifacts, such as banding in expanded regions or overexposed/underexposed areas due to color shifts naturally emerge at extreme parameter values. These provide additional learning information related to visual discomfort, even without explicitly modeling artifact types.

## 3.2 SUBJECTIVE EVALUATION

We employed a subjective pairwise comparison protocol to rank the distorted video versions. The evaluations were conducted via crowdsourcing using a side-by-side preference selection interface. In each trial, participants were presented with two videos and asked to select the one with superior visual quality. Three response options were available: "left," "right," or "can't choose."

Each participant evaluated 50 image pairs, including 5 hidden validation pairs with known preference outcomes. These validation pairs were generated by applying strong tone distortion (to overexpose or underexpose the whole picture) to introduce clearly perceptible quality degradation. All pairs were presented in randomized order, and participants were unaware of the validation samples or the underlying generation methods.

To ensure data reliability, we filtered out responses from participants who failed to correctly answer all validation pairs. Furthermore, to maintain consistency and reduce annotation bias, we balanced the pairwise comparison matrix such that each video pair received exactly 10 valid votes.

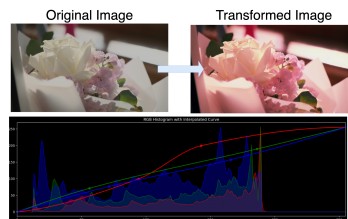

Figure 2: Example of image modifications using randomly generated tone curves. Independent curves are applied to the red, green, and blue channels, producing diverse color and contrast changes. The transformations simulate variations introduced by ITM algorithms and are used to elicit human aesthetic preferences during subjective evaluation.

Accurate and representative labeling requires displays capable of reproducing HDR content. To meet this requirement, we decided to conduct the subjective evaluation on smartphones, rather than desktop monitors. While smaller screen sizes may reduce the visibility of some spatial details, making them suboptimal for certain types of evaluations, our task focused primarily on the overall image visual attractiveness. Also modern smartphones are significantly more likely to have an HDR-capable display compared to consumer desktop monitors. To ensure that participants were viewing images on HDR displays, we required all annotators to use a dedicated mobile application. This application verified whether the participant's device supported HDR. Only users with confirmed HDR-capable screens were permitted to participate in the study. In total, we collected responses from over 3,000 crowdsourcing participants, resulting in a robust and high-coverage preference dataset.

## 3.3 ARTIFACTS LABELING

In this section, we describe the labeling of technical artifacts that frequently appear in images processed by inverse tone mapping (ITM) algorithms. Our analysis of 15 representative ITM methods applied to a set of 200 images reveals that most artifacts can be broadly categorized into two types: natural loss-based artifacts and neural network induced artifacts. Natural loss-based artifacts originate from the inherent limitations of the SDR representation compared to the original scene. These artifacts occur when ITM algorithms fail to restore or even unintentionally exaggerate missing information. Common manifestations include overexposed or underexposed regions resulting from dynamic range compression, as well as banding artifacts, which stem from insufficient bit-depth reconstruction. In HDR space, these bands become more pronounced as the luminance quantization steps increase, leading to visible discontinuities in smooth gradients. Neural network–induced artifacts arise due to imperfections in the learned representations of deep learning–based ITM models. These errors are often concentrated in highlight regions but can also appear in mid-tones or shadows. Figure 1 presents a representative example of the artifact. To construct a dataset of technical artifacts, we began with a set of 1,000 FullHD images. For each image, we applied either (a) one of the transformations described earlier (limited to tone mapping operators and tone curves), or (b) one of three neural-network based ITM methods identified in our pristine analysis as producing the highest number of neural network induced artifacts. From each transformed image, we then extracted multiple random crops, resulting in a dataset of over 100,000 image patches.

To annotate these crops, we employed crowdsourced assessment. Each crop was presented to annotators alongside the corresponding full image, and participants were asked whether the crop ex-

hibited any visible technical artifacts. Prior to evaluation, all assessors completed a brief training session that introduced and illustrated the artifact types of interest. Each participant annotated 50 image-crops, including 5 hidden validation samples with known artifact presence or absence. These validation cases were manually selected in advance. Only responses from participants who correctly labeled all validation samples were retained in the final dataset.

# 4 METHOD

As discussed earlier, assessing the quality of HDR images generated from up-converted SDR content requires consideration of two key aspects: (1) aesthetic quality, including overall tone distribution and color palette, and (2) artifact detection, targeting typical distortions introduced during the ITM process. To address both aspects, the proposed metric is designed as a dual-branch architecture, with separate modules for aesthetic prediction and artifact analysis.

The aesthetic branch operates on a downscaled version of the image, under the assumption that global aesthetic properties do not require high-resolution input. This branch is trained to predict overall visual pleasance based on tone, contrast, and color balance. The artifact branch, in contrast, operates on image patches, enabling the detection of localized distortions such as banding, overexposure, or underexposure. Patch-level scores are computed and then averaged to produce the final artifact score for the full image. The outputs of both branches are then fused using a multi-layer perceptron (MLP), which learns to combine aesthetic and technical indicators into a final unified quality score. This structure ensures that the model captures both global visual appeal and localized degradations, aligning closely with human perception of ITM-enhanced images.

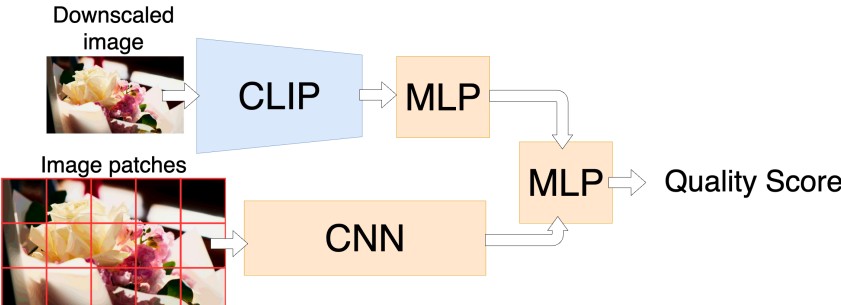

Figure 3: Overview of the proposed no-reference ITM quality assessment framework. The architecture consists of two specialized branches: an aesthetic branch, which operates on a downscaled image using CLIP features to predict overall visual pleasantness, and an artifact detection branch, which analyzes full-resolution patches to detect localized distortions.

We first describe the aesthetic branch, followed by the artifact detection branch. Each branch is trained independently to specialize in its respective task. After pretraining, their feature extraction backbones are frozen, and we train a fusion module to combine their outputs into a final quality score.

## 4.1 AESTHETIC QUALITY

In this section, we describe the aesthetic branch, which is designed to assess the overall visual pleasantness of an image from the perspective of a human observer. This component focuses on high-level perceptual attributes such as tone distribution, color harmony, and global composition.

Recent works have shown that CLIP models possess a strong emergent ability to represent visual aesthetic qualities. While CLIP was originally trained for cross-modal retrieval tasks, its image encoder learns rich, high-level semantic and compositional features that align surprisingly well with human judgments of aesthetic quality such as color harmony, balance, texture, and spatial layout.

Leveraging this observation, we adopt a frozen CLIP image encoder as the core of our aesthetic quality branch. In this branch, the input HDR image is first downscaled before being passed through

CLIP's vision encoder. This downscaling is justified by the fact that aesthetic judgments are largely based on global perception rather than fine-grained details. The CLIP embedding is then passed through a lightweight regression head, composed of two fully connected layers with ReLU activations and dropout, which outputs a scalar aesthetic quality score.

The Bradley–Terry scores obtained from subjective pairwise comparisons are inherently scaled according to the aggregation procedure, which may result in inconsistent dynamic ranges across different content. To ensure comparability and facilitate model training, we normalized all scores to the [0, 1] range using min–max normalization.

The regression head is trained using mean squared error (MSE) loss against scaled aesthetic preference scores collected in our dataset. Empirically, we find that this branch alone performs strongly on aesthetic-related tasks, validating the suitability of CLIP embeddings for perceptual quality modeling.

## 4.2 Artifacts detection module

In this section, we describe the artifact detection branch, which is responsible for identifying and quantifying localized distortions commonly introduced during the inverse tone mapping (ITM) process. While the aesthetic branch captures the global appearance of an image, it may overlook subtle or localized artifacts that significantly degrade perceived quality. To address this, we introduce a second branch dedicated to technical quality assessment through artifact detection. This module focuses on detecting technical degradations such as banding, overexposure, underexposure that may negatively impact the perceived quality of HDR images.

After a series of preliminary experiments, we adopted a pretrained model originally proposed in [ref], which was developed for banding artifact detection primarily on SDR images. Although the original domain differs from ours, we hypothesized that the model could still extract meaningful features relevant to artifact detection in HDR content, particularly given the similarity in low-level distortions such as bit-depth limitations and gradient discontinuities.

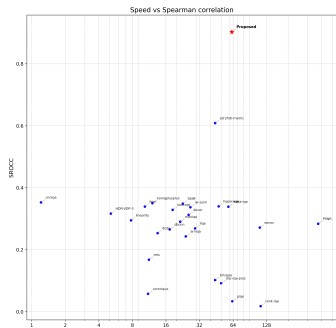

We adapted and retrained the model, a compact seven-layer convolutional neural network (CNN), to predict image patches quality. The network consists of a sequence of convolutional and pooling layers, interleaved with batch normalization, and concludes with a global average pooling layer followed by fully connected layers that output the binary classification score. Batch normalization accelerates convergence and introduces a regularizing effect, reducing overfitting on small datasets. The global average pooling layer aggregates spatial information across the receptive field, transforming the final feature maps into a compact 1D representation without introducing additional parameters.

Figure 4: Speed–quality trade-off between the proposed metric and existing quality assessment methods. The x-axis represents inference speed (in FPS), and the y-axis Spearman correlation with the subjective votes.

The model is trained using the binary cross-entropy loss, where the output scores are voted if there is an artifact on the patch. During inference, the network is applied to multiple patches extracted from the input image, and the final artifact score is computed by averaging the individual patch-level predictions. This approach enables robust detection of localized distortions while maintaining computational efficiency across full-resolution images.

This lightweight yet effective architecture allows for efficient patch-based artifact detection, which serves as the foundation for the artifact branch of our proposed ITM quality metric.

Table 1: Metrics correlations with the subjective scores.

|    | Metric | PLCC↑ | SROCC↑ | KROCC↑ | RMSE↓ |
|----|--------|-------|--------|--------|-------|
| NR | sdr2hdr-metric Voronin et al. (2024) | 0.661 | 0.609 | 0.428 | 0.271 |
|    | CNNIQA Kang et al. (2014) | 0.351 | 0.353 | 0.332 | 0.452 |
|    | KONCEPT Hosu et al. (2020) | 0.350 | 0.329 | 0.296 | 0.312 |
|    | DOVER Wu et al. (2023) | 0.340 | 0.313 | 0.300 | 0.355 |
|    | LIQE Zhang et al. (2023) | 0.335 | 0.269 | 0.312 | 0.400 |
|    | HYPERIQA Su et al. (2020) | 0.327 | 0.340 | 0.300 | 0.292 |
|    | METAIQA Zhu et al. (2020) | 0.325 | 0.339 | 0.307 | 0.325 |
|    | LINEARITY Li et al. (2020a) | 0.305 | 0.295 | 0.303 | 0.375 |
|    | KONIQ++ Su et al. (2021) | 0.300 | 0.350 | 0.357 | 0.408 |
|    | DBCNN Zhang et al. (2020) | 0.295 | 0.266 | 0.262 | 0.410 |
|    | MANIQA Yang et al. (2022) | 0.289 | 0.291 | 0.245 | 0.318 |
|    | EONSS Wang et al. (2019) | 0.278 | 0.271 | 0.260 | 0.457 |
|    | BRISQUE Mittal et al. (2012) | 0.243 | 0.102 | 0.005 | 0.596 |
|    | ARNIQA Agnolucci et al. (2024) | 0.207 | 0.243 | 0.242 | 0.413 |
|    | CLIP-IQA+ Wang et al. (2023) | 0.043 | 0.092 | 0.161 | 0.492 |
| FR | SSIM | 0.408 | 0.349 | 0.318 | 0.342 |
|    | FSIM Zhang et al. (2011) | 0.382 | 0.339 | 0.329 | 0.371 |
|    | IW-SSIM Wang & Li (2010) | 0.374 | 0.337 | 0.317 | 0.383 |
|    | HDR-VDP-3 et al. (2011) | 0.333 | 0.317 | 0.288 | 0.410 |
|    | PSNR | 0.299 | 0.284 | 0.243 | 0.470 |
|    | DISTS Ding et al. (2020) | 0.253 | 0.254 | 0.189 | 0.614 |
|    | HDR-VQM Narwaria et al. (2015) | 0.185 | 0.305 | 0.220 | 0.430 |
|    | CONTRIQUE Madhusudana et al. (2022b) | 0.158 | 0.058 | 0.032 | 0.583 |
|    | **Proposed** | **0.904** | **0.903** | **0.652** | **0.151** |
|    | **Proposed (w/o artifact detection)** | 0.891 | 0.874 | 0.618 | 0.182 |
|    | **Proposed (w/o aesthetic evaluation)** | 0.684 | 0.739 | 0.471 | 0.260 |

## 5 EVALUATION

### 5.1 COMPARISON WITH IQA/VQA MODELS

**Dataset**: For the evaluation of our proposed metric, we rely on data from the publicly available ITM benchmark Voronin et al. (2024). This benchmark comprises 20 diverse video sequences, each processed using 14 different ITM algorithms. All videos are encoded with the Hybrid Log-Gamma (HLG) transfer function, commonly used for HDR content. Subjective evaluations were conducted in a controlled setting using a MacBook Pro featuring a 1600 nit HDR display.

**Evaluation Criteria**: To evaluate the performance of the proposed quality metric, we employ four standard statistical measures: Pearson's Linear Correlation Coefficient (PLCC), Spearman's Rank-Order Correlation Coefficient (SROCC), Kendall's Rank-Order Correlation Coefficient (KROCC), and the Root Mean Square Error (RMSE). RMSE quantifies the average deviation of predicted scores from the ground truth MOS, assessing the absolute prediction error. Correlation coefficients (PLCC, SROCC, and KROCC) range from 0 to 1, with higher values indicating stronger agreement with human perception. In contrast, lower RMSE values indicate better numerical fidelity to subjective scores.

Table 1 presents a comparative evaluation of existing image and video quality assessment metrics against the proposed method. We benchmark our metric alongside widely used full-reference and no-reference general-purpose metrics, as well as a specialized HDR-oriented metric.

The results clearly demonstrate that the proposed metric significantly outperforms existing no-reference methods, which generally fail to capture the perceptual quality of images produced by SDR-to-HDR ITM algorithms. While full-reference metrics perform slightly better than generic no-reference alternatives, their effectiveness remains limited due to the inherent mismatch between ITM-generated outputs and ground truth HDR references.

Among the compared methods, the existing ITM-specific metric achieves the best results after the proposed, yet its correlation with human opinion scores remains moderate. In contrast, our proposed metric sets a new state of the art in ITM quality assessment, achieving substantially higher correlation with subjective scores. Overall, these findings highlight that our method represents a significant step forward in reliable, scalable, and perceptually aligned evaluation of ITM algorithms.

We also evaluated the runtime performance of all quality metrics, expressed in frames per second (FPS). FPS is computed as the total execution time for processing an entire video sequence divided by the number of frames in the sequence. For each metric, we report the maximum achieved FPS across all test videos Figure 4 illustrates the speed–quality trade-off between existing quality assessment metrics and the proposed method, comparing inference speed (in FPS) against Spearman rank correlation (SRCC) with human opinion scores. The proposed metric achieves a strong balance between accuracy and efficiency, placing near the median in terms of computational complexity while significantly outperforming all alternatives in prediction quality. This demonstrates the practicality of the method for real-world applications, where both perceptual alignment and runtime efficiency are critical.

## 5.2 ABLATION STUDY

To evaluate the effectiveness of each individual component, we perform an ablation study on the technical and aesthetic branches of our model to assess their respective contributions to overall performance. As shown in Table 1, both branches achieve high correlation with subjective scores when evaluated independently. However, the aesthetic branch consistently outperforms the technical branch, suggesting that perceptual factors such as tone, color balance, and overall visual appeal play a dominant role in human quality judgments.

We hypothesize that the aesthetic branch implicitly captures certain artifacts (e.g., overshooting), contributing to its strong performance. Nonetheless, the presence of visible artifacts can significantly degrade perceived quality. This is reflected in the fact that the technical branch, when combined with the aesthetic branch, enhances the overall model performance, highlighting its complementary role. Thus, both branches are essential: the aesthetic branch aligns with human preferences, while the technical branch ensures robustness in the presence of distortion.

## 6 CONCLUSION

In this work, we addressed the critical gap in evaluating the visual quality of HDR images generated through inverse tone mapping (ITM) by proposing a novel no-reference quality metric tailored specifically for this task. Existing full-reference and no-reference metrics fall short in the ITM setting due to the ill-posed nature of the problem and the perceptual divergence between restored and reference HDR content. To overcome these limitations, we constructed a large-scale, human-labeled dataset capturing multiple dimensions of visual quality, including aesthetic preference and artifact presence. Our proposed metric was trained on this dataset and achieves state-of-the-art performance, exhibiting the highest correlation with subjective human judgments across existing ITM benchmarks. We demonstrate that our method outperforms both traditional full-reference metrics and existing no-reference approaches. This work establishes a new direction for perceptual quality assessment in ITM, enabling more accurate and scalable evaluation of HDR enhancement methods, and providing essential tools for guiding future ITM algorithm development.

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
