# OpenReview forum: "hDRIVE: HDR Image Visual Evaluation Metric for SDR to HDR Upconversion Quality Assessment"
_ICLR.cc/2026/Conference — ICLR 2026 Conference Withdrawn Submission_

### Official Review · Reviewer_qovQ · 2025-10-20

**Soundness:** 1
**Presentation:** 1
**Contribution:** 2
**Rating:** 2
**Confidence:** 4

**Summary:**

The paper proposes a new no-reference metric for evaluating SDR-to-HDR inverse tone mapping (ITM) results, claiming to achieve state-of-the-art correlation with human judgments. The authors also introduce a new human-labeled dataset focusing on aesthetic preference and technical artifact perception for HDR content. The metric is a dual-branch network combining a CLIP-based aesthetic branch and a CNN-based artifact detection branch.

**Strengths:**

The paper addresses a timely and valuable problem and proposes a reasonable dual-branch design for perceptual HDR quality assessment.
However, each claimed contribution suffers from missing or inconsistent details: the image/video scope is unclear, the dataset is not accessible or reproducibly described, and HDR preprocessing and evaluation protocols are insufficiently specified.

**Weaknesses:**

1. The title and abstract conflict: the title claims an HDR image metric, while the abstract repeatedly calls it a video quality metric.
2. The method itself is image-based, yet the evaluation uses video datasets (Voronin et al., 2024). No temporal modeling, motion handling, or frame aggregation method is described.
3. The abstract states that “the metric and dataset are available for download,” but no link or preview is provided in either the paper or the supplementary materials.
4. No visualization of the dataset is shown (no tone/LUT examples, no patches, no aesthetic pair examples).
5. The annotation procedure uses mobile HDR screens without standardized luminance, tone mapping, or environmental light control, which introduces uncontrolled bias.
6. The HDR-to-CLIP preprocessing pipeline is not described. CLIP was trained on SDR sRGB data—how were HDR values mapped?
7. References include placeholders like “[ref]” and incomplete “et al.” entries.

**Questions:**

As mentioned above.

---

### Official Review · Reviewer_ou3w · 2025-10-29

**Soundness:** 1
**Presentation:** 1
**Contribution:** 2
**Rating:** 2
**Confidence:** 4

**Summary:**

In this submission, the authors try to address the critical need for a reliable quality metric for SDR-to-HDR upconversion (ITM). The authors propose HDRIVE, a no-reference, dual-branch metric trained on a new, large-scale dataset.

The architecture separates the problem into: 1) an Aesthetic Branch (a frozen CLIP encoder on a downscaled image) to predict global visual preference, and 2) an Artifact Branch (a patch-based CNN) to detect local distortions.

The dataset itself is also a major contribution, collected via pairwise comparisons on a mobile app that verified the user's display was HDR-capable. The authors promised they will make this dataset public.

**Strengths:**

1. The scale and diversity of the dataset. In terms of scale, the subjective evaluation involved gathering responses from over 3,000 crowdsourcing participants, resulting in a robust preference dataset, and the technical artifact labeling extracted over 100,000 image patches. The diversity in stimulus was achieved by generating rich and perceptually varied distortions through methods like parameterized tone mapping operators, the incorporation of over 200 3D LUTs, and automated color grading via tone curve manipulation.

2. The choice of evaluation platform and verification. I agree that evaluation on smartphones makes sense since they grow really fast for HDR content consumption. The verification step is critical to ensure the capacity of display of participants.

3. The design of dual-branch network is interesting, and the performance and efficiency of the proposed method shows a significant advantage among compared methods.

**Weaknesses:**

1. Too many unknown details.
- The specific origin or identity of the base content is unknown, such as the initial 1,000 FullHD images or the videos used for subjective preference collection.
- The sources of LUTs and color grading algorithms are unknown.
- The reference of an adopted pretrained model for artifact detection is a placeholder citation (L351).

2. The evaluation condition. Although the authors verify the participants' smartphones have an HDR display, but HDR displays differ to each other significantly. For example, an iPhone 15 Pro with a 2000-nit peak XDR display is completely different from a 3-year-old mid-range Android phone that technically supports the HDR10 standard with a 600-nit panel. Furthermore, the view condition is totally uncontrolled, and the brightness of the screen may changing according to ambient lights. Finally, there are possibly on-device tone-mapping algorithms to adapt the displayed image with the screen capacity, which are again unspecified in the dataset.

3. The validation of the method. It is unclear that how the other methods are prepared for the comparison. For trainable methods, it would make more sense to train them on the collected dataset. It seems to me the validation is mixing two perspectives of contribution together. Further, from the ablation study,  the model's performance is overwhelmingly dominated by the aesthetic branch. It would be better to try on other benchmarks to separate the evaluation of aesthetics and quality.

**Questions:**

Is it a video metric or an image metric? The entire methodology describes a 2D, static image metric (CLIP image encoder, image patch CNN) applied frame-by-frame, which is blind to temporal artifacts. However, the collected dataset and the validation one seem video datasets.

**Details Of Ethics Concerns:**

The sources of the HDR videos, images, and LUTs are unknown.

---

### Official Review · Reviewer_TUpE · 2025-10-31

**Soundness:** 2
**Presentation:** 2
**Contribution:** 2
**Rating:** 2
**Confidence:** 5

**Summary:**

This manuscript identifies a critical gap in the current evaluation of ITM algorithms, where existing full-reference and no-reference metrics fail to align well with human perception due to the ill-posed nature of the task. It constructed a large-scale human-labeled HDR images dataset generated from SDR-to-HDR upconversion and proposes a novel no-reference quality assessment metric specifically designed for evaluating HDR images.x

**Strengths:**

1. A primary contribution of this work is the creation and public release of a large-scale human-annotated dataset tailored for ITM quality assessment. This directly addresses a critical bottleneck in the field—the lack of subjectively labeled data for training and benchmarking.

2. The proposed metric is a novel, dual-branch architecture specifically engineered for the ill-posed problem of ITM quality assessment.

**Weaknesses:**

1. There are many conflicting descriptions in the manuscript. For example, the manuscript aims to design a model for video evaluation, yet the dataset construction and the computational model design appear to be for images. There is no design seen that targets video data.

2. The fusion mechanism of the proposed metric between the aesthetic and artifact branches is not described in sufficient detail.

3. While the dataset is a strength, more information on its size, diversity, and potential biases would be helpful.

4. The manuscript does not deeply discuss limitations or failure cases of the proposed metric.

**Questions:**

1.	Could you provide more details on how the outputs of the aesthetic and artifact branches are fused? Is the fusion module trained end-to-end after branch pre-training?

2.	How does the model perform on HDR content that was not generated via ITM (e.g., native HDR)? Is the metric generalizable to other HDR quality assessment tasks?

3.	Were any strategies used to ensure the dataset covers a diverse set of content and distortion types beyond the synthetic transformations?

4.	The benchmark for evaluation is limited. Could you provide more evaluation results on more HDR datasets?

5.	It should be further clarified whether there is a significant gap between the synthetic distortion techniques introduced in this manuscript for creating distorted images and the distortions caused by existing inverse tone mapping (ITM) algorithms, and whether these techniques are sufficiently capable of simulating the distortions produced by ITM algorithms.

6.	In line 225, it states, "Each participant evaluated 50 image pairs." However, the previous descriptions in this manuscript were all about video evaluation. What is the reason for this contradiction with the earlier description?

7.	Sections 3.1 and 3.3 seem to describe the construction of image datasets, while Section 3.2 describes the evaluation of video data. The author should provide a clearer explanation for this.

8.	Why construct an aesthetic dataset in Section 3.1 and an artifact dataset in Section 3.3 separately, thus requiring two large-scale subjective annotation processes? Why not construct a single dataset and annotate these several dimensions simultaneously in one annotation process?

9.	The author used an application to verify participants' devices for HDR support, but these details are not clearly described. What is the maximum brightness (luminance) of the HDR images used? What criteria must a user's display meet to be considered a "ready" HDR display? Will inconsistencies among different users' mobile phone displays cause additional color differences or brightness discrepancies? When such discrepancies exist, is calibration necessary to ensure that the same data source appears consistent to different users?

---

### Note · Authors · 2025-11-12

I have read and agree with the venue's withdrawal policy on behalf of myself and my co-authors.